# A Buzz for Sustainability and Conservation: The Growing Potential of Citizen Science Studies on Bees

**Sheina Koffler** [1,*,†] **, Celso Barbiéri** [2,†] **, Natalia P. Ghilardi-Lopes** [3] **, Jailson N. Leocadio** [4] **, Bruno Albertini** [4] **, Tiago M. Francoy** [2] **and Antonio M. Saraiva** [1,4]

1    Instituto de Estudos Avançados, University of São Paulo, R. Praça do Relógio 109,
     São Paulo 05508-970, SP, Brazil; saraiva@usp.br
2    Escola de Artes, Ciências e Humanidades, University of São Paulo, R. Arlindo Bettio 1000,
     São Paulo 03828-000, SP, Brazil; celso.barbieri@usp.br (C.B.); tfrancoy@usp.br (T.M.F.)
3    Centro de Ciências Naturais e Humanas, Federal University of ABC, R. Arcturus 3,
     São Bernardo Do Campo 09606-070, SP, Brazil; natalia.lopes@ufabc.edu.br
4    Escola Politécnica, University of São Paulo, Avenida Professor Luciano Gualberto 158, Tv. 3,
     São Paulo 05508-010, SP, Brazil; jailsonleocadio@usp.br (J.N.L.); balbertini@usp.br (B.A.)
*    Correspondence: sheina.koffler@usp.br
†    Shared co-first authorship.

**Abstract:** Expanding involvement of the public in citizen science projects can benefit both volunteers and professional scientists alike. Recently, citizen science has come into focus as an important data source for reporting and monitoring United Nations Sustainable Development Goals (SDGs). Since bees play an essential role in the pollination ecosystem service, citizen science projects involving them have a high potential for attaining SDGs. By performing a systematic review of citizen science studies on bees, we assessed how these studies could contribute towards SDG reporting and monitoring, and also verified compliance with citizen science principles. Eighty eight studies published from 1992 to 2020 were collected. SDG 15 (Life on Land) and SDG 17 (Partnerships) were the most outstanding, potentially contributing to targets related to biodiversity protection, restoration and sustainable use, capacity building and establishing multi stakeholder partnerships. SDG 2 (Zero Hunger), SDG 4 (Quality Education), and SDG 11 (Sustainable Cities and Communities) were also addressed. Studies were found to produce new knowledge, apply methods to improve data quality, and invest in open access publishing. Notably, volunteer participation was mainly restricted to data collection. Further challenges include extending these initiatives to developing countries, where only a few citizen science projects are underway.

**Keywords:** bee monitoring; beekeeping; citizen science principles; pollination; sustainable development goals

## 1. Introduction

Citizen science (CS), which can be defined as the involvement of (usually unpaid) volunteers in the scientific process (e.g., data collection, analysis, and interpretation) [1,2], has been used for different purposes, including biodiversity and environmental monitoring of both terrestrial and aquatic ecosystems. Data produced through citizen science initiatives is speeding up, in an unprecedented way, understanding of patterns and functions in biodiversity [3–5], thereby contributing towards natural resource management, environmental protection and policymaking, as well as fostering public input and engagement [6–8]. Increasingly, knowledge that is deeply integrated across disciplines and co-produced with non-academic stakeholders [9] is needed to achieve the 17 United Nations Sustainable Development Goals (SDGs), which aim for a better and more sustainable future for all by 2030, seeking the end of poverty, the improvement of health and education, the reduction of inequalities, the stimulus to economic growth, while tackling climate change and working to conserve our oceans and forests [10]. More recently, citizen science has been recognized

as a source of data for SDG reporting and monitoring, thus potentially contributing to 76 of the 244 SDG indicators [11]. Certain features of CS data are of extreme relevance, such as: spatial reference, resolution and extent; duration and temporal resolution; thematic subject areas, definitions and resolution; data purposes, use, collection, processing and management (if data is findable, accessible, interoperable and reusable); and levels of participant involvement [12]. If used in accordance with ethical and scientific principles [13,14], CS has enormous potential to expand knowledge about global biodiversity, reducing taxonomic and spatial biases in global biodiversity data sets, moving beyond data on the occurrence of single species and providing further understanding of ecological interactions among species or habitats [15].

Since they are generally small-sized and can easily fit into photographs, insects pose significant opportunities for citizen science approaches, more so than with most other biological groups. Nonetheless, apart from the possibility of sampling in many different situations, there still remain several pertinent, and as yet, unanswered scientific queries [16]. Bees are excellent subjects for mutually integrating citizen science projects and SDGs, since they comprise the most dominant pollinating taxon [17] and their ecological importance is widely recognized by the public [18]. Although bees are not the most diverse group of pollinators (as butterflies and moths, beetles, and flies show higher species richness, [17] and other pollinating insects contribute significantly to flower visitation and fruit set in crops globally [19]), bees are still the major pollinating group of wild and crop plants [20]. The global economic value of pollination has been estimated as between US$ 235 and 577 billion [20]. However, the diversity of wild and managed bees has crucial ecological, economic and social importance beyond crop pollination [21]. Indeed, bees were recognized as contributing to 15 of the 17 SDGs and at least 30 relevant SDG targets. They can easily be linked to SDGs, such as SDG 15 (Life on land), SDG 11 (Sustainable cities and communities), and SDG 12 (Responsible consumption and production) [11]. They can also be successfully used in educational programs, especially those dealing with the environment, ecology and conservation [22], directly linked to SDG 4 (Quality Education).

Here we performed a systematic review of studies combining citizen science and bees. In each case, assessment focused on addressing each study to the appropriate SDG, besides evaluating system traits, citizen participation, methods employed and research questions. Our aim was to understand the potential contribution of these particular citizen science studies to SDG reporting and monitoring, and how they conformed to ECSA principles, in order to place in evidence opportunities for enhancing practices in this field.

## 2. Material and Methods

The review carried out in this study followed the guidelines proposed by PRISMA [23], which defines a systematic review as a study employing appropriate and explicit methods to identify, select and critically evaluate relevant research through data collection and analysis.

### 2.1. Search Process

The survey of the literature was carried out in two steps: a naive search with pre-selected author terms, followed by a second step with less biased terms after using the `litsearchR` package [24]. All analyses were performed using the R version 4.0.0 (R Core Team 2020). `litsearchR`, besides automating several steps of systematic review, employs the Rapid Automatic Keyword Extraction (RAKE) algorithm to identify potential keywords initially omitted by researchers, thus improving reproducibility and reducing bias.

The searches in the literature were performed in Web of Science and Scopus databases, which returned 102 and 114 articles, respectively, during 'naive search'. The initial selection of the search terms was according to the PECO (Population, Exposure, Comparator, Outcome) framework [25]. Population was represented by "bees", Exposure by "citizen science", and Outcome by research aims and questions which were our interest in this study. Citizen science terms included those described by Eitzel et al. [26], which provide a

historical overview of terminology in CS. No Comparator terms were used. Sequentially, the `litsearchR` algorithm package was applied to this initial list of articles, according to the protocol and parameters suggested by the package authors [27]. After identifying new synonyms and related words for bees and citizen science, research was resumed on December 7 to include novel recently published articles. The final results were 122 articles in Web of Science and 134 in Scopus (total n = 256). The queries searched the terms in the title, abstract and keywords and considered all records available in the databases (the terms in italics were suggested by the `litsearchR` package):

(( "bee" **OR** "bees" **OR** "Apoidea" **OR** "Antophila" **OR** *"honey bee"* **OR** *"honeybee"* **OR** *"apis mellifera"* **OR** *"beekeep *"* **OR** *"bee colon *"* **OR** *"coloss"* **OR** *"queen problem *"* **OR** *"young queens"* **OR** *"bumble bee *"* **OR** *"bumblebee"* *"brood cell*"* **OR** *"brood comb"* **OR** *"native stingless"* **OR** *"australian stingless"* **OR** *"frieseomelitta ningra"* **OR** *"geotrigona acapulconis"* **OR** *"lestrimelitta chamelensis"* **OR** *"melipona fasciata"* **OR** *"scaptotrigona hellwegeri"* **OR** *"pot-honey"* **OR** *"african carder"*)
**AND**
( "citizen science" **OR** "participatory science" **OR** "crowd * science" **OR** "volunteer * monitoring" **OR** "networked science" **OR** "collaborative research" **OR** "collaborative monitoring" **OR** "collaborative science" **OR** *"participatory action research"* **OR** *"community action research"* **OR** *"crowdsourcing"* **OR** *"community-based participatory research"* **OR** *"local knowledge"* **OR** *"volunteered geographic information"* **OR** *"public participation in scientific research"* **OR** *"community science"* **OR** *"citizen scientist"*))

Of these, 101 were removed through duplicate analysis, six were introductions of conference proceedings (and were also removed), and 149 remained as potentially relevant study items. In addition, 13 others were selected through citation in the revised studies or author personal knowledge, resulting in a total of 162 studies. Our search strategy was as inclusive as possible, in an effort to include studies that fulfilled the established requirements, even when the term "citizen science" was not present. This is especially important, since an assessment of ornithological studies showed that many studies employing CS data did not explicitly mention volunteer participation in data collection [28], which may also have interfered in our search results.

*2.2. Collected Data*

Titles and abstracts were read to identify studies that did not conform to the requirements for analysis, e.g., could not be characterized as CS, did not include bees (only other pollinators), or which only mentioned or recommended CS, but did not include citizen scientists in any step of procedures. Following Eitzel et al. [26] recommendations, we only considered as citizen science those studies in which volunteers were actively involved in some aspect of the project and were informed how their data was going to be used. Thus, data gathered at online databases (without the owners' knowledge or consent) and, thus not related to CS programs, were not considered. In addition, studies regarding local knowledge assessments were only included if participants were knowledge-producers (not study subjects, e.g., Smith et al. [29]). After screening, 74 articles were removed and 88 remained for review (Supplementary Material 1). A subset of 40 articles were read, each by two researchers, to validate and standardize the terms and categories used for each indicator up for analysis (Table 1). The remainder were distributed equally between them. All articles were analyzed, and the most prominent UN SDG related to each one was defined. SDG 2 (Zero Hunger) was related to studies of the influence of bees on agricultural production, SDG 4 (Quality Education) to those in which bees were used to promote scientific education (this process was the main focus of the study), SDG 11 (Sustainable Cities and Communities) to studies of the biology of bees on urban landscapes, SDG 15 (Life on Land) to those aimed at investigating biological or ecological aspects of bees in general,

such as species identification, occurrence and distribution, and their interaction with plants, among others, and SDG 17 (Partnerships) to studies in which establishing a partnership, such as recruitment, engagement, retention strategies or co-creation of platforms, or validating a citizen science project, the case of pilot-testing of protocols, and data quality analysis, were central to research. SDGs are related to distinct targets, each measured by distinct indicators. Hence, we also indicated which targets could be addressed and their indicator tier classification [30]. Indicators are classified into three tiers, tiers I and II having established methods and standards. However, for tier II indicators data collection is not regular countrywide. Currently, the global indicator framework does not include any tier III indicators, which are those with no methods and standards available yet.

**Table 1.** Variables assessed in each study retrieved in the systematic review. Details are given for the variable name, information source (A: article or P: project), definition and levels considered for each variable, and which ECSA principle was being assessed (see also Table S2).

| Group | Variable Name | Source | Definition | ECSA Principle |
|---|---|---|---|---|
| Study information | Proponents (adapted from [31]) | A | Affiliation of study authors | - |
| | Funding source (adapted from [31]) | A | Funding institution cited/acknowledged in the article | - |
| | Reach | A | Spatial scale (local, regional, or global) | - |
| | Country | A | Country or countries where the project was performed | - |
| | Project duration | P | Project length and status (short or long-term, active or finished) | - |
| | SDG | A | UN Sustainable Development Goal addressed by the study (see explanation in the text) | - |
| | Research subject | A | Research areas explored (when more than one aim was declared, only the main results were considered) | 9 |
| | Hypothesis-led | A | Whether authors clearly state that there is a study question or hypothesis to test | 2 |
| | Data quality (modified from [32]) | A | Strategies employed to improve data quality in the citizen science projects | 3, 6 |
| Project information | Project name | P | Citizen science project name | - |
| | Project purpose (modified from [2]) | P | Aim of volunteer participation in the project (biological recording, biological monitoring, crowd-sourcing, or creating technology platforms) | - |
| | Degree of participation (modified from [33]) | A | Whether study was contributory, collaborative, co-created, or used crowd-sourced data. | 1, 3, 4 |
| Studied system | Animal group | A | Which animal groups were studied | - |
| | Taxon | A | Taxonomic name of the focused group | - |
| | Sociality | A | Sociality level of the bees studied | - |
| Participant information | Number of participants | A | Number of participants contributing to the project. Descriptive statistics were based on exact numbers provided by the study (approximations were not considered) | - |
| | Volunteer profile | A | Profile of participants. "General public" was inferred when no other profile was mentioned | - |
| | Recruitment | A | Methods employed for recruiting participants | - |
| | Communication | A | Methods and tools used to train participants and deliver relevant information about the project | 3 |
| | Volunteer assessment (what?) (modified from [34]) | A | Learning outcomes and perceptions of volunteers | 3 |
| | Volunteer assessment (how?) | A | Mechanisms for volunteer assessment | 3 |
| | Volunteer assessment (when?) | A | At which step volunteers were assessed (pre/post survey) | 3 |
| Ethics commitment | Open access | A | Publication type regarding accessibility or whether study was a conference paper | 7 |
| | Feedback to participants | A | Whether authors mention if any feedback was given to the volunteers | 5 |
| | Acknowledgements | A | Whether citizen scientists were acknowledged in the study. | 8 |

Additionally, the compliance with ECSA principles [13] was analysed for each article, as follows: 1. CS projects actively involve citizens in scientific endeavour that generates new knowledge and understanding; 2. CS projects should have a genuine scientific outcome. 3. Both professional scientists and citizen scientists benefit from taking part; 4. Citizen scientists may, if they so wish, participate in multiple stages of the scientific process; 5. Citizen scientists receive feedback from the project; 6. CS is considered a research approach like any other, with limitations and biases that should be considered and controlled; 7. CS project data and metadata are made publicly available and where possible,

results are published in an open access format; 8. Citizen scientists are acknowledged in project results and publications; 9. CS programs are evaluated for their scientific output, data validity, participant experience, and wider societal and policy impact; and finally, 10. The leaders of CS projects should take into consideration the legal and ethical issues surrounding copyright, intellectual property, data sharing agreements, confidentiality and attribution, as well as the environmental impact of any activity (see Table 1 for the correspondence of each variable and ECSA principle and Table S2 for detailed criteria used for each principle). Even though distinct aspects are considered in each individual principle, compliance with the principle was treated as a binary variable, which means that when more than one indicator was proposed, we considered the principle as fulfilled if at least one of them was contemplated in the study. Not all aspects covered by the ECSA Principles could be assessed in our analysis because they were not explicitly mentioned in the published results. We highlight here the difficulty to evaluate all aspects of the 9th ECSA principle, because of the intrinsic complexity for the measure of project outputs and impact [14]. The 10th principle was not evaluated because a deep analysis of each project would be necessary since the information presented in the manuscripts does not always include the legal and ethical aspects of the CS initiatives in detail. Despite these limitations, we believe our analyses provide an interesting framework for systematically assessing CS research, which may be further expanded in future studies.

## 3. Results

Of all the 88 scientific studies that were collected, 81 were peer-reviewed articles and seven conference papers. From 1992 to the present, there has been a constant increase in the number of studies published per year, reaching a maximum of 19 in 2019 (Figure S1). Most studies were undertaken on a local scale (Figure 1) and comprised data from long-term citizen science projects (73%, n = 60), including 47 ongoing projects.

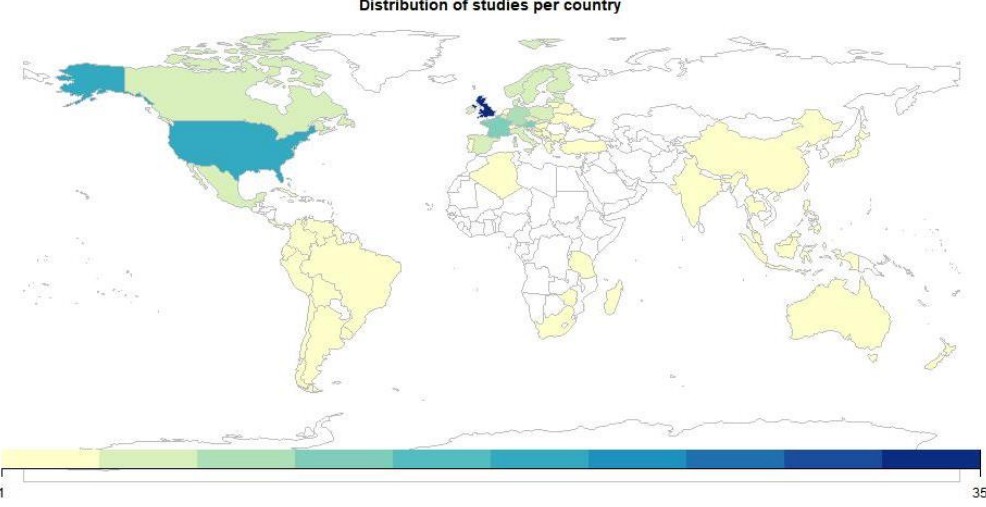

**Figure 1.** Global distribution of citizen science studies of bees. All the reported countries were gathered for each study. The study was not represented when individual countries were not reported in the article ( e.g., "Europe"). The map was constructed using the `'rworldmap'` R package [35].

### 3.1. Sustainable Development Goals

The studies were mainly addressed to SDG 15 (Life on Land, 52.3%), followed by SDG 17 (Partnerships, 29.5%), with the remainder to SDGs 2, 4, and 11 (Zero Hunger, Quality Education, and Sustainable Cities and Communities, respectively—Figure 2a). Even though the reviewed studies were not explicitly related to any SDG, data and findings produced could be used to monitor and implement 12 tier I and 9 tier II indicators (Table S3). Main research subjects were beekeeping, distributional ecology (with 6 studies focusing on invasive species), data quality, natural history, plant-pollinator interactions, volunteer

assessment, and landscape ecology (Figure 2b). Few studies focused on population ecology, agricultural practices, toxicology, or were descriptions of new projects. Project purposes were biological recording (63.9%), biological monitoring (30.7%), providing technology platforms (2.3%), and crowd-sourcing (1.1%).

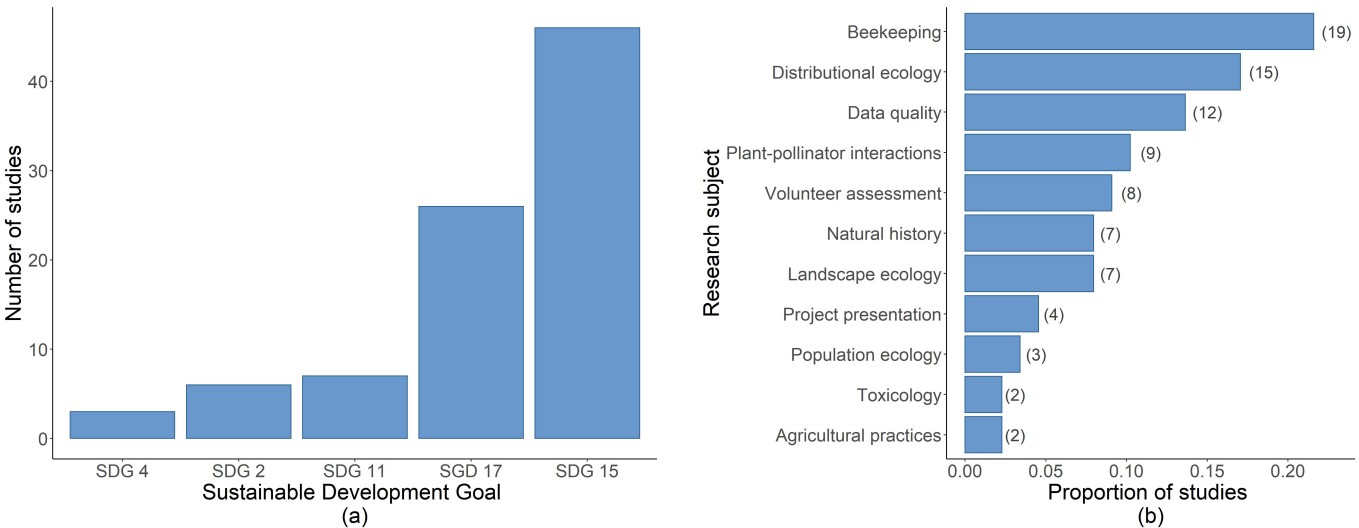

**Figure 2.** (**a**) Absolute number of studies related to each SDG identified in the review process and (**b**) Proportion of research subjects expressed in the studies analysed (number of studies is represented after the bar).

Bees were the only study system in 76% of the studies, whereas 24% included other animals, such as insects, invertebrates, or birds (complete dataset available in Supplementary Material 1). Among those, 65.9% focused on social bees, 21.6% on both social and solitary species, and 12.5% only on solitary species. Most studies gathered data for *Bombus* spp. (35.2% with focus on the genus level or the specific species), and honey bees (*Apis mellifera*, 26.1%. Stingless bees were only investigated in three studies. While honey bee studies were frequently related to beekeeping (83%), bumble bee studies showed a more diverse pattern of research subjects. Solitary bee studies, on the other hand, were usually related to distributional ecology. Several studies (18%) were based on data provided by long-term citizen science projects on these bee groups, such as Bee Watch, Bumble Bee Watch, and COLOSS (COLOSS survey of honey bee colony losses).

Funding was mainly provided by governments (68.2%), followed by non-governmental organizations (NGOs, 28.4%), universities (22.7%), and the private sector (13.6%). While two studies declared no funding to report or personal funding, ten made no mention (Figure S2a). University members were authors in 93.2% of the studies, while governments and NGOs were represented in 39.8% and 38.6%, respectively. Three studies also had authors from the private sector and one from a school (Figure S2b). The number of participants in each project varied from 2 to 28,629 (Figure S3); however 32% of the studies do not mention how many participants were involved. Although in several cases (36.4%) there was no mention of how participants were recruited, the most frequent manner was through digital media (38.6%), or related organizations (43.2%), such as beekeeper or gardener associations. Participant communication and training was either online (51.8%), through presencial meetings and workshops (22.9%), or through manuals (24.1%).

### 3.2. ECSA Principles

The reviewed papers scored points on a scale from 2 to 9 (Figure 3). All the reviewed papers scored points for the principles 1 and 2. We considered these principles classificatory to be included in the analysis. The average score of the 88 reviewed papers is 6.7, the minimum 3 (1.1%) and the maximum 9 (7.9%). A total of 38 papers scored 7 points (43.2%). Excluding the principles 1 and 2, the most frequent principles were Principle 4

(97.7%) and Principle 6 (94.3%), while the less frequent ones were Principle 9 (35.2%) and Principle 5 (23.9%).

**Figure 3.** Compliance to ECSA principles (**a**) Number of studies according to total score (number of principles fulfilled) and (**b**) Proportion of studies following each ECSA principle.

When considering the first and second ECSA principles, notably hypotheses or scientific questions were explicitly stated in 50% of the studies. Although several (48.8%) relied on volunteers from the general public, without targeting any specific group, beekeepers, bee enthusiasts, and students were target groups in some. Considering volunteer participation (fourth ECSA principle), data collection was the main task performed by citizen scientists, and 93.2% of the studies were classified as contributory. Two studies were collaborative, three co-created, and one relied on crowd-sourcing. From all selected studies, 37.5% contained information on feedback to participants (fifth ECSA principle) and 53.4% were published as open access (seventh ECSA principle). Volunteers were acknowledged in 79.5% of the studies (eighth ECSA principle), and two article included participants as authors [29,36].

Few studies (22%) included some kind of volunteer assessment (ninth ECSA principle) and those that did, focused on interest, motivation, behavior, knowledge and perception. Assessment, applied through questionnaires, was generally applied after volunteer participation (Table S1). As to data quality evaluation (ninth ECSA principle), different strategies were employed, with a maximum of five per study (Figure 4). Digital vouchers (photographs submitted by citizen scientists), expert review of data, use of structured protocols, and training of participants were the most frequent strategies to improve data quality. Volunteer personal knowledge, usually related to beekeepers' experience in analyzing hive conditions, was also exploited to improve data quality.

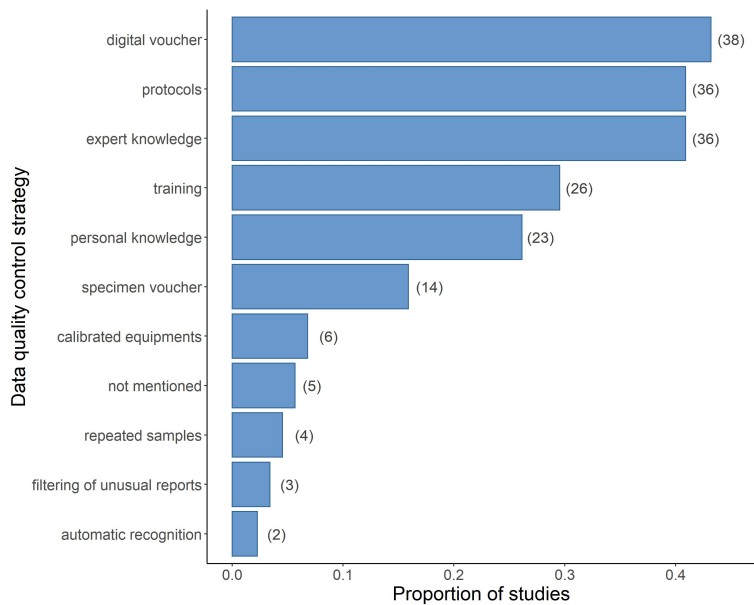

**Figure 4.** Proportion of data quality control strategies employed in citizen science studies of bees. Each study could apply more than one strategy. Number of studies is represented after the bar.

## 4. Discussion

Our results revealed that five of the 17 SDGs could be addressed by citizen science studies of bees. In general, the aims were to expand basic bee knowledge, investigate applied aspects of beekeeping and conservation, and explore the potential of CS as a research and educational process. The recent increase noted in the number of published works follows a more general trend in scientific publications in this field, and indicates the acceptance of citizen science by the research community towards mainstreaming this research avenue [37]. Publishing results from citizen science projects in peer-reviewed journals and conferences also fulfills the first and second citizen science principles, as citizen participation is generating new knowledge and understanding, while at the same time producing genuine scientific outcomes [13]. In addition, the growing number of studies is contributing to the generation of large scale data that can be employed as a non-traditional data source for SDG monitoring and implementation [12]. In a previous analysis, bees were related to 30 targets across SDGs 1 and 15, demonstrating a great potential to contribute to SDGs achievement [21]. However, SDGs 16 and 17 were not taken into consideration by Patel et al. [21], as these goals focus on governance and policy, which nevertheless can be targeted through CS. Another positive factor is the increasing recognition by the public that bees are important providers of pollination services [38], a possible motivation for recruiting volunteers for participation in citizen science projects. In fact, the main motivational factors declared in a survey with citizen scientists from a bee monitoring program were contributing to scientific data collection and aiding in bee conservation [39].

Since bees are the most dominant pollinators of flowering plants [17] and bee diversity contributes to increasing crop production [40], studies combining agriculture and bees were related to SDG 2 (Zero Hunger). Specifically, CS projects may contribute to ensuring sustainable agriculture and increasing productivity (target 2.4). In this context, as bee monitoring in crop fields is a potential tool for sustainable agriculture, this could be implemented in collaboration with agriculture stakeholders [41]. For instance, the citizen science studies reviewed here focused on assessing data quality of bee monitoring protocols in crop fields [42], as well as management strategies for assuring adequate pollination services for food production [43].

SDG 4 (Quality Education) is related to articles regarding CS projects for schools. Studies thus addressed would contribute to relevant and effective learning outcomes,

education for sustainable development, and teacher qualification (targets 4.1, 4.7, and 4.c, respectively). Even though SDG targets and indicators focus on literacy and numeracy in primary and secondary education, we highlight the importance of scientific literacy for sustainability comprehension and implementation, which can be positively impacted by participation in citizen science projects [44,45]. Regarding vocational and technical skills, Patel et al. [21] also suggest that training for beekeeping may provide equitable opportunities for men, women, and indigenous people possessing traditional knowledge, which can be considered as an innovative opportunity for citizen science projects in SDG 4.

SDG 11 (Sustainable Cities and Communities) was explored when investigating bee response to urbanization, a key conservation factor when considering the growing urban sprawl. Reviewed studies regarding this SDG mainly focused on landscape ecology, hence related to protecting and safeguarding the world's natural heritage and providing access to green and public spaces (targets 11.4 and 11.7). For instance, CS data revealed foraging resource availability was related to solitary bee nesting [46], and how the proportion of impervious surface affected bee communities [47]. Thus, there is a positive relationship between bee presence and gardens, urban green spaces and the remaining natural ecosystems within cities, which in turn, benefit from bee pollination. None of the studies focused on employing bees for air quality monitoring in cities, another potential application for citizen science and SDG 11 [21].

Expanding knowledge of basic bee biology and ecology contributes to SDG 15 (Life on Land), potentially promoting conservation and the sustainable use of terrestrial ecosystems. CS projects provide large-scale spatial and temporal data, allowing estimating species distribution and assessing extinction risk [48–50]. These results may ultimately be useful for identifying potential areas for biodiversity protection and informing national conservation policies, hence attaining targets 15.1, 15.4, 15.5, and 15.9. Regarding alien species (target 15.8), the high sampling effort in citizen science projects could facilitate recording and monitoring invasive processes [51,52]. Furthermore, as bees are important pollinators [17], data obtained by citizen scientists on flower visitation and pollination outcome [53,54], could ultimately contribute to conservation and restoration programs that rely on effective pollination (targets 15.1–15.5). Despite the increasing number of CS studies on bees, there is a bias for social species, especially honey bees and bumblebees. Solitary bees, which comprise the majority in bee diversity [55], are largely unknown by the public [38], contrary to what occurs with managed bees. Thus, gaps in bee diversity are still found in CS research, with few projects focusing on native solitary and stingless bees, which exhibit high species richness resulting in difficulties in species identification by non-experts. This is especially important in tropical countries, where species diversity is high. All told, partnerships with beekeepers [56] could provide an opportunity for the sustainable use of biodiversity, as well as fair and equitable sharing of benefits from the use of genetic resources (targets 15.1 and 15.6).

Establishing partnerships (SDG 17) is essential for successful CS projects. Inducing capacity building and multi stakeholder partnerships, and involving the various social sectors that are central elements in citizen science projects, are in line with targets 17.9, 17.16, and 17.17. In this respect, our results corroborate the findings of Cunha et al. [31], showing that governments and NGOs play major roles in building and funding partnerships, whereas in the private sector this is less so. Private sector participation (17.17) could lead to inducing companies to support science and sustainability, hence rewarded with innovations in technology and production (SDG 9 Industry Innovation and Infrastructure). On the other hand, as study proponents are mainly researchers, this could affect the intended role of each stakeholder. Indeed, the level of participation in most of the studies reviewed was contributory, thereby indicating that citizen scientist participation is restricted to collecting or processing data [33]. According to the fourth principle of citizen science, citizen scientists may, if they so wish, participate in several stages of the research process [13]. Similar to previous assessments, contributory approaches were more frequent [2]. Even though requiring more effort and engagement from both researchers

and citizen scientists, involving higher levels of citizen participation in collaborative and co-created projects is also relevant as they promote greater ownership and may bring contributions driven by the needs of the community that may be related to key SDG targets [12]. The scarcity of co-created projects may be explained once one of the main disadvantages in CS approaches is related to the amount of effort and difficulty to execute a protocol or participate in the project. The co-created and collaborative projects usually need more effort from the professional scientists to maintain the volunteers committed and manage the research, because the data acquisition is not under the professional team direct control [57].

According to Martens [58], spatial scale is an essential factor in sustainable development. Most of the reviewed studies concentrated on a local scale, which contributes to monitoring volunteers, their activities and expectations [57]. In addition, citizen science projects contributing to monitoring at the local scale have greater potential to implement SDGs in specific contexts, and if successful, feasible for scaling up when attempting to reach a global level [12]. Scaling CS initiatives to the global level may contribute to bee monitoring programs, which can provide essential information on how pollinators face global change [59]. Indeed, participatory research has been increasingly indicated as a powerful strategy for long-term pollinator monitoring, suggesting an avenue for mainstreaming CS in bee and pollination research and advocating for funding those initiatives [60–62]. Since the Global Biodiversity Information Facility (GBIF) already accepts CS data, including these data on research will become more common and highlights the importance of data sharing practices. Currently and according to previous studies, most projects were carried out in developed countries, mostly in Europe and North America [5,63]. The United Kingdom stands out as a leader in citizen science projects involving bees, which was to be expected, due to the UK's centenary tradition in biodiversity monitoring projects [64]. On the contrary, in developing countries, where access to research funding is more restricted and levels of formal education lower, there are less citizen science projects and added difficulties in volunteer engagement [63,65]. Investing in citizen science projects on developing countries as a strategy to achieve SDGs is highly relevant because these countries concentrate the greatest and most important biodiversity hotspots, are responsible for the maintenance of numerous ecosystem services, and show high levels of poverty and inequality [66]. Citizen science thus emerges as a promising way to engage local communities in conservation projects, besides being a potentially cheaper way to monitor biodiversity, especially important in developing countries. Furthermore, citizen science initiatives focused on beekeeping should be supported in developing countries, because this activity can be a promoter of sustainability in its social, economic, cultural and ecological domains [22], being valuable to reduce social and gender inequalities [21].

Regarding best practices and the use of correct terminology, during our search several studies using the term "citizen science" were found. However, they were not considered as CS in our study, according to current definitions and so were excluded from our analyses [26]. Even though the term "citizen science" was found in keywords of some studies, authors only mentioned or suggested CS. Two allegedly CS studies used data collected from people stored in social media platforms, not involving citizen scientists on the science making process, and being in conflict with the ECSA principles. The distinction between this kind of data mining and the crowdsourced level of participation in citizen science must be highlighted, because consent is an ethical principle of this research field. The term citizen science should be carefully used in scientific publications, once this field is a valid scientific approach like any other with their own strengths and limitations and the appropriate use of terminology contributes to its recognition as a field of study [26].

Volunteer assessment was rarely explored in the studies, thereby hampering an evaluation of outcomes from the volunteer's point of view (third principle, [13]). Knowledge of the volunteer's motivations is crucial for keeping them engaged, especially in the case of long-term citizen science projects [57]. However, we highlight that the analyzed citizen science projects may present volunteer assessment protocols, even though this feature was not exploited in the publications. Results from citizen scientist outcomes are highly

valuable when designing new or improving long-term projects [67]. Recruitment and communication strategies employed in citizen science projects also exert great influence when engaging citizen scientists [57]. Recruiting a particular profile volunteer with a close relationship with the research subject, may be a simple manner of improving data quality, since the volunteer can rely on personal knowledge to run the protocols [32]. This strategy was used in several of the studies surveyed here, where beekeepers were recruited in large numbers through already existing social organizations [54,56]. Volunteer participation was also assessed as to the quality of data, always a major concern in citizen science studies [12,32]. In most of the cases, the focus was on reliability in species identification, still a challenge in citizen science projects on biodiversity. Strategies to overcome taxonomic uncertainty may involve data validation by experts, identification restricted to higher-level groups or non-natural groups, and focusing on easily identifiable species [68,69].

Feedback to volunteers on the research is of utmost importance for maintaining citizen scientist motivation and collaboration throughout the project [70]. More so, communicating project outcomes represents an ethical principle for professional researchers (fifth and tenth principles [13]). Furthermore, high quality science communication is essential not only for spreading specific knowledge, but also building up trust between the population and scientific community [71]. Although there was mention of the various communication strategies, several did not mention feedback to participants, possibly since this feature was unrelated to the research aim. Nonetheless, numerous studies were published as open access, and most acknowledged volunteers, thus in accordance to the seventh and eighth principles of citizen science [13]. Even though open access publishing in citizen science is still hampered by elevated costs [72], citizen science articles on bees were more frequently published in this format (56%), in comparison with statistics for general publication (20.4% [73]). Notwithstanding, volunteer acknowledgement should be strongly recommended, seeing that in 22% of the studies there was none.

## 5. Conclusions

Since sustainability can only be achieved through an intergenerational approach [58,74], establishing firm partnerships is an essential step, whereby citizen science can act as a powerful strategy for providing data and implementing SDGs [12]. Our findings revealed that existing citizen science projects are already contributing to scientific research, and even though none were directly aligned to SDGs, studies implicitly related to and data derived from these projects are linked to 21 indicators from 18 SDG targets. Among these, nine are tier II indicators without regular data production, and thus can be positively impacted by citizen science projects contributing with data [12].

Nevertheless, citizen science research on bees still presents major gaps, such as the lack of volunteer assessment, which would significantly contribute to building efficient volunteer engagement strategies, improving learning outcomes, and promoting meaningful experiences. To include citizen science in the SDG workflow and implementation in local contexts, these issues should be dealt with. The inclusion of citizen scientists in all the steps of the scientific process is still uncommon, and should be fostered in further studies. Major challenges are citizen science in developing countries, where investments on research are constrained and budget cuts frequent [75,76]. Few citizen science papers have been produced in developing countries, especially in the southern hemisphere. Thus, these represent priority areas for formulating participatory and co-created projects aiming at achieving several SDGs related to bees, beekeeping, and biodiversity.

**Supplementary Materials:** The following are available online at https://www.mdpi.com/2071-105 0/13/2/959/s1, Supplementary Material 1: Complete dataset on citizen science studies assessed in this systematic review; Supplementary Material 2: Figure S1: Publication of citizen science studies on bees from 1992 to 2020, Figure S2: Funding source and proponent affiliation of the citizen science studies on bees, Figure S3: Number of participants engaged in the citizen science studies on bees, Table S1: Variables assessed in the systematic review and summarized results, Table S2: Compliance

criteria for the ECSA Principles, Table S3: SDG targets and indicators related to the citizen science studies on bees.

**Author Contributions:** Conceptualization, S.K. and C.B. methodology, S.K., C.B., N.P.G.-L., J.N.L., B.A. and T.M.F.; validation, S.K., C.B., N.P.G.-L., J.N.L., B.A. and T.M.F.; formal analysis, S.K. and C.B.; investigation, S.K. and C.B.; data curation, S.K. and C.B.; writing—original draft preparation, S.K., C.B., N.P.G.-L., J.N.L. and T.M.F.; writing—review and editing, S.K., C.B., N.G., J.N.L., T.M.F. and B.A.; visualization, S.K., C.B., J.N.L. and B.A.; supervision, N.P.G.-L., B.A., T.M.F. and A.M.S.; project administration, A.M.S., T.M.F., N.P.G.-L. and B.A.; funding acquisition, A.M.S. All authors have read and agreed to the published version of the manuscript.

**Funding:** This research was funded by Fundação de Amparo e Apoio à Pesquisa do Estado de São Paulo (FAPESP, grant numbers 2018/14994-1 and 2019/26760-8). This study was financed in part by the Coordenação de Aperfeiçoamento de Pessoal de Nível Superior—Brazil (CAPES)—Finance Code 001; C.B. grant number 88882.377160/2019-01; J.L grant number 88882.333367/2019-01, and by Conselho Nacional de Desenvolvimento Científico e Tecnológico—Brazil (CNPq), A.S. grant number 312605/2018-8.

**Institutional Review Board Statement:** Not applicable.

**Informed Consent Statement:** Not applicable.

**Data Availability Statement:** Complete datasets are provided as Supplementary Material.

**Acknowledgments:** We thank all citizen scientists and researchers from the studies reviewed here for their contribution to bee conservation and sustainability. We also would like to acknowledge researchers from the SURPASS2 project (Safeguarding pollination services in a changing world) for their support to our study. The SURPASS2 project is funded under the Newton Fund Latin America Biodiversity Programme: Biodiversity—Ecosystem services for sustainable development, awarded by the UKRI Natural Environment Research Council (NERC: NE/S011870/2), in partnership with the Argentina National Scientific and Technical Research Council (CONICET 1984/19), Brazil/São Paulo Research Foundation (FAPESP 2018/14994-1), and Chile National Agency for Research and Development (ANID NE/S011870/1).

**Conflicts of Interest:** The authors declare no conflict of interest.

## Abbreviations

The following abbreviations are used in this manuscript:

| | |
|---|---|
| SDG | Sustainable Development Goals |
| CS | Citizen Science |
| ECSA | European Citizen Science Association |
| PRISMA | Preferred Reporting Items for Systematic Reviews and Meta-Analyses |
| RAKE | Rapid Automatic Keyword Extraction |
| PECO | Population, Exposure, Comparator, Outcome |
| NGO | Non-Governmental Organizations |
| GBIF | Global Biodiversity Information Facility |

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
