# Peer review of "A Buzz for Sustainability and Conservation: The Growing Potential of Citizen Science Studies on Bees"

_sustainability, doi:10.3390/su13020959_

Round 1

Reviewer 1 Report

The goal of this paper is to provide a systematic review of studies combining citizen science and bees. Based on this analysis, the authors seek to:

  1. identify and prioritize areas where citizen science bee research is being used to inform the UN Sustainable Development Goals indicators (Tier I and II), and where it could be used to inform the SDG indicators;
  2. evaluate the general performance of citizen science bee projects in relationship to the European Citizen Science Association's 10 Principles of Citizen science, which outlines ethical best practices. 

Overall, this paper has the potential to make a big impact, but there are some major revisions that are needed. 

Major comments:

Achieving the first goal would make a substantial contribution to the field. The authors are able to identify several areas where citizen science bee research may contribute to the SDGs, and to some degree, they were able to assess the performance of the projects with respect to the ECSA principles. However, the conflation of these two goals and the organization of the paper greatly undermined the potential impact the paper otherwise might have had. My strong recommendation would be to revise the paper, focusing primarily on contributions of citizen science bee research to the SDGs, and then perhaps develop a follow up paper focused on performance in relation to the ECSA principles. 

The paper would benefit from an introduction and brief literature review that introduces the SDGs, why they are important, and how bee research might inform the SDGs generally. Then, describe why citizen science bee research may offer unique insights for the SDGs that without citizen science would not be possible. For example, Tier I and II indicator standards are already well established. Where might citizen science bee research contribute to the development of Tier III indicators, to move them into Tier II and I? 

In the body of the paper, there is limited discussion of how citizen science bee research has impacted bee research -- what new discoveries have emerged using citizen science that would not have been possible otherwise? Also, how has citizen science bee research informed environmental and resource management decisions / policies that advance the SDGs? Was this discussed in the literature that was analyzed? See, for example, for comparison the literature on e-Bird that makes direct connections between the research and policy change/impact. 

Lastly, the authors note that many or most of the projects surveyed focused on local contexts. Perhaps the authors could comment also on the opportunities to scale citizen science bee research from local to national to regional to global. How would that impact the science understanding of bees, of climate, etc?

Minor Comments:

  • Cooper et al discovered that many scientific papers do not specifically call out the use of citizen science, although they may use citizen science data. It would be good to mention this limitation in your analysis.
  • There are at least 15 terms in academia for citizen science like approaches, and CSTP journal published a good article that describes the range of terms. The terms used for this paper under review are a subset. This limitation might be noted in the paper. Question - which terms do the bee researchers tend to use?
  • A few more tables and graphs would have helped with the readability of the paper. 
  • Use subheaders to break the paper up into more digestible sections, and also coalesce ideas. The rapid alternating between SDGs and ECSA principles throughout the document diluted the impact and made it harder to follow. 
  • The authors make a statement on page 8 that few projects used "crowdsourcing", however, the authors did not search using the term "crowdsourcing" when they searched for papers to review.
  • The authors show a bit of bias towards co-created projects. There is a place for contributory projects, and a place for co-created projects, and the choice between them really should depend on the scientific or societal goals that need to be achieved. One approach is not necessarily better or worse than the other. It depends on the fit. 

Author Response

The goal of this paper is to provide a systematic review of studies combining citizen science and bees. Based on this analysis, the authors seek to:

  1. identify and prioritize areas where citizen science bee research is being used to inform the UN Sustainable Development Goals indicators (Tier I and II), and where it could be used to inform the SDG indicators;

  2. evaluate the general performance of citizen science bee projects in relationship to the European Citizen Science Association's 10 Principles of Citizen science, which outlines ethical best practices. 

Overall, this paper has the potential to make a big impact, but there are some major revisions that are needed. 

> We would like to thank the referee for the constructive review. We have followed the suggestions and believe our manuscript has improved significantly.

Major comments:

Achieving the first goal would make a substantial contribution to the field. The authors are able to identify several areas where citizen science bee research may contribute to the SDGs, and to some degree, they were able to assess the performance of the projects with respect to the ECSA principles. However, the conflation of these two goals and the organization of the paper greatly undermined the potential impact the paper otherwise might have had. My strong recommendation would be to revise the paper, focusing primarily on contributions of citizen science bee research to the SDGs, and then perhaps develop a follow up paper focused on performance in relation to the ECSA principles. 

> We have reorganized all the sections of our paper in two subsections, the first regarding SDGs and the second ECSA Principles, in order to improve clarity. In addition, we have improved the systematic assessment for ECSA principles, also following a suggestion made by the reviewer 2. We are now presenting which variable was explicitly used to assess each principle, allowing a more quantitative approach on the evaluation of the principles followed by citizen science studies.

The paper would benefit from an introduction and brief literature review that introduces the SDGs, why they are important, and how bee research might inform the SDGs generally. Then, describe why citizen science bee research may offer unique insights for the SDGs that without citizen science would not be possible. For example, Tier I and II indicator standards are already well established. Where might citizen science bee research contribute to the development of Tier III indicators, to move them into Tier II and I? 

> SDGs aims and its relationship with citizen science was more broadly explored in the introduction. We have altered the text as follows:

Increasingly, knowledge that is deeply integrated across disciplines and co-produced with non-academic stakeholders (Irwin et al., 2018) is needed to achieve the 17 United Nations Sustainable Development Goals (SDGs), which aim a better and more sustainable future for all by 2030, seeking the end of poverty, the improvement of health and education, the reduction of inequalities, the stimulus to economic growth, while tackling climate change and working to conserve our oceans and forests (UN, 2020). More recently, citizen science has been recognized as a source of data for SDG reporting and monitoring, thus potentially contributing to 76 of the 244 SDG indicators (Fraisl et al. 2020). ”

Citizen science in SDG reporting and monitoring is also explored in the introduction and the relationship between SDGs and bee studies is also discussed. As mentioned in the methods section, currently, UN has no tier III indicators, as all proposed indicators exhibit established methods and standards”. Potential indicators that could be addressed by CS studies on bees and their tier classification are presented in Table S3.

Indicators are classified into three tiers, tiers I and II having established methods and standards. However, for tier II indicators data collection is not regular countrywise. Currently, the global indicator framework does not include any tier III indicators, which are those with no methods and standards available yet.”

In the body of the paper, there is limited discussion of how citizen science bee research has impacted bee research -- what new discoveries have emerged using citizen science that would not have been possible otherwise? Also, how has citizen science bee research informed environmental and resource management decisions / policies that advance the SDGs? Was this discussed in the literature that was analyzed? See, for example, for comparison the literature on e-Bird that makes direct connections between the research and policy change/impact. 

> Even though CS is a recent approach in bee research, its impacts in conservation and policy are increasingly being debated in the literature. In addition, CS is recommended for long-term monitoring of pollinators, which we believe will be employed more frequently in the future. The revised studies in our work do mention the importance of CS in informing environmental and resource management decisions and policies, however, we have not found a direct example of the application of CS studies so far. We highlight this importance as follows:

Indeed, participatory research has been increasingly indicated as a powerful strategy for long-term pollinator monitoring, suggesting an avenue for mainstreaming CS in bee and pollination research and advocating for funding those initiatives (Dicks et al. 2016, Harvey et al. 2020, Science for Environment Policy 2020).”

Lastly, the authors note that many or most of the projects surveyed focused on local contexts. Perhaps the authors could comment also on the opportunities to scale citizen science bee research from local to national to regional to global. How would that impact the science understanding of bees, of climate, etc?

> The potential for scaling up CS research and how this would affect understanding of bee responses to global changes was further discussed in our study. The following sentences were included in the discussion section:

Scaling CS initiatives to the global level may contribute to bee monitoring programs, which can provide essential information on how pollinators face global changes (Potts et al. 2016). Indeed, participatory research has been increasingly indicated as a powerful strategy for long-term pollinator monitoring, suggesting an avenue for mainstreaming CS in bee and pollination research and advocating for funding those initiatives (Dicks et al. 2016, Harvey et al. 2020, Science for Environment Policy 2020). Since the Global Biodiversity Information Facility (GBIF) already accepts CS data, including these data on research will become more common and highlights the importance of data sharing practices.”

Minor Comments:

  • Cooper et al discovered that many scientific papers do not specifically call out the use of citizen science, although they may use citizen science data. It would be good to mention this limitation in your analysis.

> We acknowledge that scientific studies may not use the term “citizen science” explicitly. To reduce this limitation, we have tried to perform an inclusive search strategy by employing several related terms, using the litsearchR protocol to avoid researcher bias, and including studies that we have found while revising the literature or that were already known by the authors. We discuss this further in the methods section and also refer to Cooper et al. work:

In addition, 13 others were selected through citation in the revised studies or author personal knowledge, resulting in a total of 162 studies. Our search strategy was the most inclusive as possible, in an effort to include studies that fulfilled the established requirements, even when the term “citizen science” was not present. This is especially important, since an assessment of ornithological studies showed that many studies employing CS data did not explicitly mention volunteer participation in data collection (Cooper et al. 2014), which may also have interfered in our search results.”

  • There are at least 15 terms in academia for citizen science like approaches, and CSTP journal published a good article that describes the range of terms. The terms used for this paper under review are a subset. This limitation might be noted in the paper. Question - which terms do the bee researchers tend to use?

> We acknowledge this limitation and have tried to reduce it by including other CS terms identified by Eitzel et al. (2017, CSTP) in our systematic search. 

Citizen science terms included those described by Eitzel et al. (2017), which provide a historical overview of terminology in CS.”

Indeed, broadening our search has improved our findings, resulting in an inclusion of seven new studies to our dataset. These studies included research on local knowledge. however studies where the participants were the study subject were not included (i.e. scientific knowledge being produced by participants was the criterion used to include the study in our dataset). In addition, by reviewing the terms identified by Eitzel et al. (2017), we have restricted the criteria for recognizing the study as CS and now distinguish crowdsourcing (CS) from data-mining (not CS).

Two allegedly CS studies utilized data collected from people stored in social media platforms, not involving citizen scientists on the science making process, and being in conflict with the ECSA principles. The distinction between this kind of data mining and the crowdsourced level of participation in citizen science must be highlighted, because consent is an ethical principle of this research field.”

Citizen science” was the most frequent term found in the reviewed studies, however we have also found few records for “community science”, “crowdsourcing”, “participatory science”, “public outreach”, “citizen participation” and “local ecological knowledge”.

  • A few more tables and graphs would have helped with the readability of the paper.

> We have included a figure to represent ECSA principle compliance by the studies (Figure 3), which we believe has improved understanding of our results. 

  • Use subheaders to break the paper up into more digestible sections, and also coalesce ideas. The rapid alternating between SDGs and ECSA principles throughout the document diluted the impact and made it harder to follow. 

> Thank you for your suggestion, we have divided our manuscript in two sections as recommended.

  • The authors make a statement on page 8 that few projects used "crowdsourcing", however, the authors did not search using the term "crowdsourcing" when they searched for papers to review.

> We have included the term “crowdsourcing” in our systematic search to avoid any bias. This has resulted in the inclusion of one new study to our dataset.

The authors show a bit of bias towards co-created projects. There is a place for contributory projects, and a place for co-created projects, and the choice between them really should depend on the scientific or societal goals that need to be achieved. One approach is not necessarily better or worse than the other. It depends on the fit. 

> We have corrected this bias, as we agree that both contributory and co-created projects are valuable and have distinct goals. The following sentences were included:

Similar to previous assessments, contributory approaches were more frequent (Pocock et al. 2017). Even though requiring more effort and engagement from both researchers and citizen scientists, involving higher levels of citizen participation in collaborative and co-created projects is also relevant as they promote greater ownership and may bring contributions driven by the needs of the community that may be related to key SDG targets (Fritz et al. 2019). The scarcity of co-created projects may be explained once one of the main disadvantages in CS approaches is related to the amount of effort and difficulty to execute a protocol or participate in the project. The co-created and collaborative projects usually need more effort from the professional scientists to maintain the volunteers committed and manage the research, because the data acquisition is not under the professional team direct control (Pocock et al 2014).”

Reviewer 2 Report

The manuscript ‘A buzz for sustainability and conservation: the growing potential of citizen science studies on bees’ is a review of the current literature on Citizen Science studies in bee research. It aims also at understanding if Citizen Science bee projects can help to reach the Sustainable Development Goals by the UN. In its literature survey it identified 83 publications reaching from1992 to 2020, which are concered with bee research using the tool of Citizen Science. The authors describe the aims, tools and research species of the publications. They connect the publications with five sustainable development goals, whereas they found most frequently connections with SDG17 and SDG15. Furthermore, the quality of CS projects is evaluated by comparing them with the ECSC principles. The authors conclude that many reviewed studies help to target and reach the mentioned SDGs, but that there is a potential for improvement in implementing the ECSC principles.

The manuscript is concerned with an increasing trend in the natural sciences: to engage citizen scientists into high quality research. In this paper the focus is set on a small, but very prominently discussed group: the bees (such as honey bees, bumble bees and solitary bees). It tackles the question, if such projects can help to change the world to the better and to increase live quality of the human population, which is very relevant in times of climate change and biodiversity loss. The other aspect, comparing the project to the ECSC principles, is important to help Citizen Science research reach a high credibility in the science community.

I see this manuscript as an valuable summary of the recent research effort. It also suggests necessary improvements in research quality and research areas (e.g. using these tools also in developing countries). However, it has to be revised carefully, to increase the clarity of the manuscript. I also suggested some ideas, which could broaden the topics of the manuscript and therefore increase it’s significance.

Comments:

The author state several times, that ‘bees are the main/primary pollinators of flowering plants’ to connect bee research with the SDG 2. This statement is exaggerated and should be phrased more carefully. Bees are important pollinators, but other insects such as hoverflies, butterflies and beetles are similarly important (as even the paper cited by the author claims: Ollerton 2017 Annu. Rev. Ecol. Evol. Syst.; but see also Rader et al 2016 PNAS). I suggest the author focus more on the obvious reasons, why bees are being connected that strongly with pollination of agricultural plants: that they are the only pollinating insects, we can breed and control at large scale. And that bees are a good topic for the Citizen Science approach because of their high popularity.

This leads to the general question: why do you explicitly use bees as topic and not pollinators/insects in general? There are several Citizen Science projects for butterflies for example. Including other insect pollinators could increase the significance of the study significantly.

P2, R31-43: this is simply copy/paste from the European Citizen Science Association, although correctly cited. That seems strange for me. Please find another solution.

P2, R44-50: the authors describes why Citizen Science is a good tool for insect research, but it would be interesting to add more aspects, also negative ones. Examples for other positive aspects: high popularity of bees, some groups are relatively easy identified on photographs (bumblebees!); Negative aspects: difficulty of identification at species level for many solitary bees, low knowledge of insect groups in the human population (can be increased with projects!), under-reporting of ‘common species’, data collection not evenly distributed.

P2, R-50-52: The percentage of plants only pollinated by bees seem very high (see also general comment above), especially because the reference gives only numbers from Brazil. Please compare also with Klein et al 2007 Proc R Soc B.

P2, R54-58: the authors name here SDGs, which are not mentioned later on, and don’t mention all of the SDGs of results and discussion. Please adjust this.

P2, R59-64: At the moment the authors state ‘Our aim was to understand how…citizen science studies contributed to SDG’. In my understanding, these studies did not contribute to any reporting. That should be rephrased or clarified.

The authors state, that they test if publications conform with ECSA principles. However, these comparisons are mentioned unsystematically throughout the results and discussion. It would help to have a graph similar to Fig 2 a, demonstrating which principles are met more often than other. This could also help to avoid the copy/page paragraph from page 2, as the principles could be summarized in buzzwords in the graph. It would be also interesting to know, how many goals are met by the different publications. Just a though: could one rate the quality of the CS projects by such tests?

P2 R70 which R version is used? Please also cite the R Core Team.

P3, R86-89: why are no keywords for solitary bees in the term list? The list is very strongly centred on literature of social bees, especially the terms suggested by the R package.

P3, R108: ‘in urban landscapes’

Fig 1: it would be nice to have a better labelled legend, maybe in steps of 5?

P6, R139-142: it could be interesting, to correlate bee group and study subject. I would expect to find the honeybee mainly connected with topics such as beekeeping and plant-pollinator interaction, while bumblebees and solitary bees would be connected with the ecological topics. The same could be interesting for the SDGs.

P7 R172: ‘bee keeping’ instead of ‘bee rearing’

P8, R217-220: important point, and important to increase projects for solitary bees. In the sense: people only protect, what they know. Citizen science projects could be a good tool for this. By the way, the authors do not mention another group of social bees, stingless bees, which are tropical bees. As the authors are Brazilians, it could be interesting to point towards study deficits on the own continent (if there aren’t any stingless bee projects).

P9 R291-298: These sentences do not fit into the conclusion. Please rephrase or use them in the introduction or discussion.

Author Response

The manuscript ‘A buzz for sustainability and conservation: the growing potential of citizen science studies on bees’ is a review of the current literature on Citizen Science studies in bee research. It aims also at understanding if Citizen Science bee projects can help to reach the Sustainable Development Goals by the UN. In its literature survey it identified 83 publications reaching from1992 to 2020, which are concered with bee research using the tool of Citizen Science. The authors describe the aims, tools and research species of the publications. They connect the publications with five sustainable development goals, whereas they found most frequently connections with SDG17 and SDG15. Furthermore, the quality of CS projects is evaluated by comparing them with the ECSC principles. The authors conclude that many reviewed studies help to target and reach the mentioned SDGs, but that there is a potential for improvement in implementing the ECSC principles.

The manuscript is concerned with an increasing trend in the natural sciences: to engage citizen scientists into high quality research. In this paper the focus is set on a small, but very prominently discussed group: the bees (such as honey bees, bumble bees and solitary bees). It tackles the question, if such projects can help to change the world to the better and to increase live quality of the human population, which is very relevant in times of climate change and biodiversity loss. The other aspect, comparing the project to the ECSC principles, is important to help Citizen Science research reach a high credibility in the science community.

I see this manuscript as an valuable summary of the recent research effort. It also suggests necessary improvements in research quality and research areas (e.g. using these tools also in developing countries). However, it has to be revised carefully, to increase the clarity of the manuscript. I also suggested some ideas, which could broaden the topics of the manuscript and therefore increase it’s significance.

> We would like to thank the referee for the constructive review. We have followed the suggestions and believe our manuscript has improved significantly.

Comments:

The author state several times, that ‘bees are the main/primary pollinators of flowering plants’ to connect bee research with the SDG 2. This statement is exaggerated and should be phrased more carefully. Bees are important pollinators, but other insects such as hoverflies, butterflies and beetles are similarly important (as even the paper cited by the author claims: Ollerton 2017 Annu. Rev. Ecol. Evol. Syst.; but see also Rader et al 2016 PNAS). I suggest the author focus more on the obvious reasons, why bees are being connected that strongly with pollination of agricultural plants: that they are the only pollinating insects, we can breed and control at large scale. And that bees are a good topic for the Citizen Science approach because of their high popularity.

This leads to the general question: why do you explicitly use bees as topic and not pollinators/insects in general? There are several Citizen Science projects for butterflies for example. Including other insect pollinators could increase the significance of the study significantly.

> We have restated some sentences to describe more accurately the position of bees as the most dominant pollinator taxa, following results by Ollerton et al. (2017).  In addition, we also refer to other important pollinators as Lepidoptera, Coleoptera, and Diptera, as well as their importance in crop pollination (Rader et al. 2016). By restricting our systematic search to bees, we directed our efforts to these most frequent and more studied pollinators, which also allowed a more complex analysis of each study. The following sentences were included:

Bees are excellent subjects for mutually integrating citizen science projects and SDGs, since they comprise the most dominant pollinating taxon (Ollerton 2017) and their ecological importance is widely recognized by the public (Wilson et al. 2017). Although bees are not the most diverse group of pollinators (as butterflies and moths, beetles, and flies show higher species richness, Ollerton et al. 2017) and other pollinating insects contribute significantly to flower visitation and fruit set in crops globally (Rader et al. 2016), bees are still the major pollinating group of wild and crop plants (IPBES, 2016).”

P2, R31-43: this is simply copy/paste from the European Citizen Science Association, although correctly cited. That seems strange for me. Please find another solution.

> We have relocated these sentences to the methods section to describe in detail each principle. This citation was maintained in the manuscript since we believe it is essential for the comprehension of the following analysis of compliance to the ECSA principles.

P2, R44-50: the authors describes why Citizen Science is a good tool for insect research, but it would be interesting to add more aspects, also negative ones. Examples for other positive aspects: high popularity of bees, some groups are relatively easy identified on photographs (bumblebees!); Negative aspects: difficulty of identification at species level for many solitary bees, low knowledge of insect groups in the human population (can be increased with projects!), under-reporting of ‘common species’, data collection not evenly distributed.

> We discuss negative aspects and limitations of CS research in the discussion sections, referring to data quality and the challenges to bee identification, especially for solitary species and stingless bees. The following sentence was included:

Thus, gaps in bee diversity are still found in CS research, with few projects focusing on native solitary and stingless bees, which exhibit high species richness resulting in difficulties in species identification by non-experts.”

P2, R-50-52: The percentage of plants only pollinated by bees seem very high (see also general comment above), especially because the reference gives only numbers from Brazil. Please compare also with Klein et al 2007 Proc R Soc B.

> We have removed these values, since they only refer to crop production in Brazil. Main references now include Ollerton 2017 and IPBES 2016.

P2, R54-58: the authors name here SDGs, which are not mentioned later on, and don’t mention all of the SDGs of results and discussion. Please adjust this.

> We have changed these sentences in the introduction and now discuss the relationship between bees and SDGs more broadly citing the work by Patel et al. (2020), which reviews SDG targets that could be addressed by bee studies beyond plant pollination and food production. The following sentences were added:

However, the diversity of wild and managed bees has crucial ecological, economic and social importance beyond crop pollination (Patel et al., 2020). Indeed, bees were recognized as contributing to 15 of the 17 SDGs and at least  30 relevant SDG targets. They can also be successfully used in educational programs, especially those dealing with the environment, ecology and conservation (Barbieri et al. 2020), directly linked to SDG 4 (Quality Education).”

P2, R59-64: At the moment the authors state ‘Our aim was to understand how…citizen science studies contributed to SDG’. In my understanding, these studies did not contribute to any reporting. That should be rephrased or clarified.

> This sentence was rephrased accordingly:

Our aim was to understand the potential contribution of these particular citizen science studies to SDG reporting and monitoring, and how they conformed to ECSA principles, in order to place in evidence opportunities for enhancing practices in this field.”

The authors state, that they test if publications conform with ECSA principles. However, these comparisons are mentioned unsystematically throughout the results and discussion. It would help to have a graph similar to Fig 2 a, demonstrating which principles are met more often than other. This could also help to avoid the copy/page paragraph from page 2, as the principles could be summarized in buzzwords in the graph. It would be also interesting to know, how many goals are met by the different publications. Just a though: could one rate the quality of the CS projects by such tests?

> Thank you for your suggestion. We have performed a more systematic assessment for ECSA principles and now indicate which variable was explicitly used to assess each principle. This change allowed a more quantitative approach on the evaluation of the principles followed by citizen science studies. Results of this analysis are shown in Figure 3.

(...) (see Table 1 for the correspondence of each variable and ECSA principle and Table S2 for detailed criteria used for each principle). Even though distinct aspects are considered in each individual principle, compliance with the principle was treated as a binary variable, which means that when more than one indicator was proposed, we considered the principle as fulfilled if at least one of them was contemplated in the study. Not all aspects covered by the ECSA Principles could be assessed in our analysis because they were not explicitly mentioned in the published results. We highlight here the difficulty to evaluate all aspects of the 9th ECSA principle, because of the intrinsic complexity for the measure of project outputs and impact (Robinson et al. 2018). The 10th principle was not evaluated because a deep analysis of each project would be necessary since the information presented in the manuscripts does not always include the legal and ethical aspects of the CS initiatives in detail. Despite these limitations, we believe our analyses provide an interesting framework for systematically assessing CS research, which may be further expanded in future studies.”

P2 R70 which R version is used? Please also cite the R Core Team.

> R version 4.0.0 was used. This information was included in our manuscript.

P3, R86-89: why are no keywords for solitary bees in the term list? The list is very strongly centred on literature of social bees, especially the terms suggested by the R package.

> The search term “bee” already returns results on solitary bees and including specifically “solitary bee” does not change the output of our systematic search. The addition of new search terms related to citizen science, as suggested by Reviewer 1, resulted in a new suggested term for a solitary bee (the african carder bee) after running litsearchR.

P3, R108: ‘in urban landscapes’

 > Thank you for your careful revision, we have corrected accordingly.

Fig 1: it would be nice to have a better labelled legend, maybe in steps of 5?

> We have included more levels in Figure 1 label, however, the general trend is maintained.

P6, R139-142: it could be interesting, to correlate bee group and study subject. I would expect to find the honeybee mainly connected with topics such as beekeeping and plant-pollinator interaction, while bumblebees and solitary bees would be connected with the ecological topics. The same could be interesting for the SDGs.

> Even though we have an extremely biased dataset to honey bees and bumblebees, when relating bee group to research subject, we find that honey bees were mainly related to beekeeping studies, while bumblebees showed distinct research subjects. The same analysis was not performed regarding SDGs, since the dataset was biased to SDGs 15 and 17 and no pattern was observed. The following sentence was included:

While honey bee studies were frequently related to beekeeping (83%), bumble bee studies showed a more diverse pattern of research subjects. Solitary bee studies, on the other hand, were usually related to distributional ecology. ”

P7 R172: ‘bee keeping’ instead of ‘bee rearing’

> The term was corrected accordingly.

P8, R217-220: important point, and important to increase projects for solitary bees. In the sense: people only protect, what they know. Citizen science projects could be a good tool for this. By the way, the authors do not mention another group of social bees, stingless bees, which are tropical bees. As the authors are Brazilians, it could be interesting to point towards study deficits on the own continent (if there aren’t any stingless bee projects).

>The inclusion of new search terms in our systematic review resulted in an addition of two studies focusing on stingless bees. One study previously revised investigated stingless bees and honey bees (Castilhos et al. 2019). We include this finding in the results section and discuss the challenges to include both native solitary and stingless bee studies:

Despite the increasing number of CS studies on bees, there is a bias for social species, especially honey bees and bumblebees. Solitary bees, which comprise the majority in bee diversity (Michener 2000), are largely unknown by the public (Trip et al. 2020), contrary to what occurs with managed bees. Thus, gaps in bee diversity are still found in CS research, with few projects focusing on native solitary and stingless bees, which exhibit high species richness resulting in difficulties in species identification by non-experts.”

P9 R291-298: These sentences do not fit into the conclusion. Please rephrase or use them in the introduction or discussion.

> These sentences were moved to the discussion section as recommended.

Round 2

Reviewer 1 Report

The improvements added significant value to the paper.  This is an exciting and important area of research, and I look forward to seeing how these authors develop this research track in the future. 

Author Response

The improvements added significant value to the paper.  This is an exciting and important area of research, and I look forward to seeing how these authors develop this research track in the future. 

> We are very pleased with the final version of our manuscript and excited to continue our research on citizen science and its impacts on conservation and sustainability. We thank the reviewer for the constructive comments and suggestions, which have substantially improved our work.